# Optimized Peppermint Essential Oil Microcapsules Loaded into Gelatin-Based Cryogels with Enhanced Antimicrobial Activity

**DOI:** 10.3390/polym15132782

**Published:** 2023-06-22

**Authors:** Didem Demir, Gulden Goksen, Seda Ceylan, Monica Trif, Alexandru Vasile Rusu

**Affiliations:** 1Department of Chemistry and Chemical Process Technologies, Vocational School of Technical Sciences at Mersin Tarsus Organized Industrial Zone, Tarsus University, Mersin 33100, Türkiye; 2Department of Food Technology, Vocational School of Technical Sciences at Mersin Tarsus Organized Industrial Zone, Tarsus University, Mersin 33100, Türkiye; 3Department of Bioengineering, Faculty of Engineering, Adana Alparslan Turkes Science and Technology University, Adana 01250, Türkiye; 4Centre for Innovative Process Engineering (CENTIV) GmbH, 28857 Syke, Germany; 5CENCIRA Agrofood Research and Innovation Centre, Ion Meșter 6, 400650 Cluj-Napoca, Romania; rusu_alexandru@hotmail.com

**Keywords:** gelatin, chitosan, peppermint essential oil, microcapsule, cryogel, antibacterial

## Abstract

In this study, chitosan (Chi) was used to microencapsulate peppermint essential oil (PEO). A novel gelatin-based cryogel loaded with PEO microcapsules was further developed and characterized for potential applications. Four different cryogel systems were designed, and the morphological, molecular, physical and antibacterial properties were investigated. Additionally, the antimicrobial properties of PEO, alone and microcapsulated, incorporated into the cryogel network were evaluated. The observed gel structure of cryogels exhibited a highly porous morphology in the microcapsules. The highest values of the equilibrium swelling ratio were acquired for the GelCryo–ChiCap and GelCryo–PEO@ChiCap samples. The contact angle GelCryo–PEO@ChiCap sample was lower than the control (GelCryo) due to the water repelling of the essential oil. It has been found that the incorporation of encapsulated PEO into the cryogels would be more advantageous compared to its direct addition. Moreover, GelCryo–PEO@ChiCap cryogels showed the strongest antibacterial activities, especially against *Staphylococcus aureus* (Gram-positive bacteria) and *Escherichia coli* (Gram-negative bacteria). The system that was developed showed promising results, indicating an improved antibacterial efficacy and enhanced structural properties due to the presence of microcapsules. These findings suggest that the system may be an appropriate candidate for various applications, including, but not limited to, drug release, tissue engineering, and food packaging. Finally, this system demonstrates a strategy to stabilize the releasing of the volatile compounds for creating successful results.

## 1. Introduction

Polymer-based materials under the name of “gel” are “polymer solvent” systems in which macromolecules form a 3D network with non-fluctuating bonds. They are relatively soft materials, and are generally robust when subjected to large deformations. While the morphology of the gels is determined using the fabrication method, the nature of the bonds is determined according to the nature of the polymers used. On the other hand, the chosen solvent prevents the collapse of the system and creates a 3D mesh network rather than a compact polymer mass [1]. In general, gels can be divided into two main groups: chemical and physical, according to the nature of the intermolecular bonds at the polymer network junctions. When we want to customize them even more, as described in the review by Lozinsky et al., gels are divisible according to the physicochemical causes of the gel formation as chemotropic (intermolecular chemical bonds), ionotropic (ion-exchange reactions), chelatotropic (chelating reactions), solvotropic (changes of solvent composition), thermotropic (heating of polymers), psychrotropic (chilling of polymers), and cryotropic gels (freezing of polymers) [1].

Cryotropic gels, commonly known as cryogels, are interlinked porous polymeric substrates fabricated as a result of the crytropic gelation (cryogelation) of monomers/polymers in a suitable solvent under freezing temperatures [2,3]. The cryogelation process consists of three steps: solution preparation, freezing, and thawing. These stages can be summarized as follows: (i) First, polymers or monomers as polymer precursors (with or without a crosslinking agent and an initiator) are dissolved in a suitable solvent. (ii) The prepared homogeneous solution is poured into a mold so that the final product to be formed can be shaped and incubated below the freezing point of the solvent. During the freezing process, the solvent freezes and forms interconnected ice crystals. In the unfrozen part, the polymers/monomers polymerize or crosslink to form a network around the ice crystals. (iii) After the reaction is complete, the frozen gel is brought to room temperature and removed from the mold. Meanwhile, the ice crystals acting as pore formers melt, and the polymerized/crosslinked portions around the crystals remain unmelted and form the pore walls [4,5]. The formed gel is washed with large volumes of water to remove the unreacted ingredients. Because they are highly porous materials, they are not suitable for drying at room temperature. Generally, the drying process is carried out in a freeze dryer [6] or vacuum oven [7]. Due to their interconnected porous structure, large surface area, high swelling ability, availability in various shapes, relatively easy preparation, and short diffusion path, cryogels are increasingly being investigated for use in different areas, such as tissue engineering applications (scaffolds for tissue regeneration/formation [8], drug delivery systems [9], wound dressings [10]), environmental science (disinfection [11], filtration [12], wastewater treatment [13]), and food technology (enzyme immobilization [7], food preservation [14]). Unitl now, various kinds of monomers and polymers have been used to obtain cryogel structures. As examples of monomers, acrylate derivatives are often used with ammonium persulfate [15] and tetramethylethylenediamine [16] initiators. On the other hand, when the polymers are evaluated, chitosan, gelatin, cellulose, pectin, silk fibroin, alginate, and collagen can be provided as the most used polymers with natural origins, and poly(ethylene glycol), poly(L-lactic acid), and poly(vinyl alcohol) in the synthetic polymer class [17].

Cryogel materials have gained significant attention in biomedical and biotechnological applications due to their unique combination of physicochemical properties and biological activity. The term “biologically active” refers to the ability of these materials to interact with biological systems and elicit specific biological responses. Three-dimensional cryogel structures are produced using natural polymers mainly composed of polysaccharides, proteins, and nucleic acids produced by microorganisms, and this structure is the survival mechanism of bacteria. Therefore, it is very important to improve the antibacterial properties of these unique materials. This can be achieved by using polymers with self-antibacterial properties, as well as antimicrobial substances (such as metallic nanoparticles, carbon nanomaterials, glass/ceramics, antibiotics, and bioactive molecules) for the cryogel structure. The biological activity of cryogel materials in these applications is closely linked to their physicochemical properties, such as pore size, surface chemistry, mechanical strength, and biocompatibility. By tailoring these properties, cryogels can be designed to exhibit specific biological responses, making them versatile and promising materials in the field of biomedicine and biotechnology [18,19].

From this point of view, we aimed to improve the antibacterial properties of gelatin-based cryogels by adding peppermint essential oil (PEO), which contains bioactive molecules such as menthol, menthone, neomenthol, and iso-menthone [20]. Gelatin has been studied by many researchers to date, and it is used safely for different applications, such as food and feed, environment, and biomedical use, due to its good biocompatibility [21,22,23]. Gelatin alone is not suitable for creating a stable structure, so it can be crosslinked with glutaraldehyde to form a 3D cryogel network, as performed in previous studies [24,25,26]. On the other hand, PEO is known for its good antimicrobial and antioxidant activity and other pharmacological activities such as antiviral and antihelmentic effect [27], making it an effective candidate for food, pharmaceutical, cosmetic, and biomedical applications. To date, few articles have addressed the incorporation of PEO into polymers for specific applications such as food packaging and wound dressing materials. Huang et al. added PEO to electrospun gelatin nanofibers for potential edible packaging application [28], and Moriera et al. combined PEO with gelatin and starch for edible coating, postharvest quality, and the shelf life of guava fruits [29]. For biomedical applications, Unalan et al. studied PEO loaded on poly(ε-caprolactone) electrospun fiber mats for wound healing applications [30]. In these studies, PEO was added directly into the polymers in different amounts before the fabrication process. Alternatively, in our study, we aimed to encapsulate PEO with the help of chitosan, which is also known for its highly antimicrobial activity [31], and then add the encapsulated oil into the cryogel matrix to provide a prolonged release of the essential oil and thus extend its antibacterial activity. Plain gelatin, PEO-loaded gelatin, microcapsule-incorporated gelatin, and PEO microcapsule-incorporated gelatin cryogels were produced and compared in order to examine the effect of the bioactive addition method on the material properties and the antibacterial activity. The produced gels were characterized in terms of morphology, chemical structure, surface properties, swelling ability, and antibacterial activity.

## 2. Materials and Methods

### 2.1. Materials

Chitosan with a medium molecular weight, gelatin for microbiology, 25% (*v*/*v*) glutaraldehyde solution, glacial acetic acid, and absolute ethanol were purchased from Sigma Aldrich, St. Louis, MO, USA. PEO was obtained from a local herbal market in Mersin, Türkiye. Pure sodium hydroxide (NaOH) pellets, tryptone soy broth (TSB) and plate count agar (PCA) were supplied from Merck, Darmstadt, Germany. All chemicals were of analytical grade. Distilled water was used for preparing the solutions, dilutions, and washing steps.

### 2.2. Fabrication of Cryogels

#### 2.2.1. Gelatin (GelCryo) and PEO-Incorporated Gelatin (PEO@GelCryo) Cryogels

Gelatin-based cryogel was synthesized with some modifications, according to the described methods in our previous studies [25,32,33]. Briefly, the calculated amount of gelatin was dissolved in distilled water at 60 °C under stirring to obtain a 6% wt. gelatin solution. After a homogenous solution was obtained, 150 µL glutaraldehyde (3%, *v*/*v*) was added into the gelatin solution rapidly while stirring at room temperature. The solution was immediately poured into 1.0 cm diameter cylindrical plastic tubes to fill the mold, and put in the freezer to start forming ice crystals that act as porogens. After reaction at −20 °C in a refrigerator for 24 h, the product was thawed at room temperature, removed from the tube, and washed several times with high volumes of distilled water. The GelCryo was obtained after freeze-drying for 24 h (Figure 1A).

The PEO-loaded cryogel was also prepared by crosslinking the gelatin solution prepared at the same concentration, with the same amount of glutaraldehyde, under the same conditions. The only difference was that 1.5 times the amount of polymer (0.12 g) and PEO were added to the gelatin solution prepared before crosslinking. The product prepared at this stage was named PEO@GelCryo (Figure 1B).

#### 2.2.2. Plain Chitosan (ChiCap) and PEO-Loaded Chitosan (PEO@ChiCap) Microcapsulates

The beads were prepared by dispersing the polymer droplets in a concentrated NaOH solution. To do this, chitosan was chosen as the main polymer to prepare ChiCap and PEO-loaded PEO@ChiCap microcapsules. For this, 2% wt. chitosan solution was prepared in 4% (*v*/*v*) acetic acid solution. The homogeneously prepared solution was carefully placed in a plastic syringe with a 21 gauge needle, and dripped at a constant rate into the 10% wt. NaOH solution. The droplets were broken up into mono-dispersed droplets (1 droplet/second) by the shear force of the continuous flow in the NaOH solution. The resulting beads were soaked in the NaOH solution for at least 4 h to stabilize their shape. The beads were washed with distilled water until the washing water became neutral (Figure 1C).

To prepare the PEO@ChiCap microcapsules, 2 mL of the 2% wt. chitosan solution was taken and mixed with 0.12 g of PEO. The mixture was stirred at a constant speed for at least 1 h to ensure that the oil was completely dispersed in the polymer. The beads were formed by dripping the obtained mixture into a 10% wt. NaOH solution with the help of an injector, as mentioned above (Figure 1D).

#### 2.2.3. ChiCap-Loaded Cryogel (GelCryo–ChiCap) and PEO@ChiCap-Loaded Cryogel (GelCryo–PEO@ChiCap)

In the preparation of microcapsules-loaded gelatin cryogels, both ChiCap and PEO@ChiCap microcapsules produced in the previous step were dispersed in a gelatin solution and then crosslinked (Figure 1E). For this, microcapsules were added in a 6% wt. gelatin solution, mixed at a low speed, and dispersed into a homogeneous solution. To this mixture, 0.15 µL of 3% (*v*/*v*) glutaraldehyde solution was added, poured into cylindrical plastic tubes to take on the shape, and left overnight at −20 °C for crosslinking. The frozen gels were thawed at room temperature, washed to remove unreacted residues, and freeze-dried. In the continuation of the study, ChiCap-loaded cryogels were named GelCryo–ChiCap, and PEO@ChiCap-loaded cryogels PEO@ ChiCap–GelCryo.

### 2.3. Characterization of Microcapsules

The morphologies of the microcapsules were examined using an optical microscope (Eclipse ME600, Nikon, Tokyo, Japan). The average diameter of the beads was determined from optical microscopy images by analyzing about 50 particles using Image J software with version 1.53. The manual mode of the Image J analyzer was used for the measurement of the average diameter of the beads. Randomly selected beads were analyzed for both the long and short axes. The presence of PEO after encapsulation with chitosan was confirmed by using the FTIR spectra (Thermo Scientific, Nicolet TM IS TM, Waltham, MA, USA). The microcapsules were scanned within the frequency region of 400 to 4000 cm^−1^.

### 2.4. Characterization of Cryogels

#### 2.4.1. Chemical Structure

The chemical groups of cryogels were analyzed by using infrared (IR) spectra (Thermo Scientific, Nicolet TM IS TM, USA). Each cryogel specimen was scanned within the frequency region of 400 to 4000 cm^−1^, and the characteristic peaks of IR transmission spectra were recorded.

#### 2.4.2. Morphology

Scanning electron microscopy (FEI, Quanta 650 FEG, Eindhoven, The Netherlands) was used to analyze the surface and inner morphology of cryogels. Before imaging, the samples were coated with a thin layer of platinum. SEM was used at the acceleration of 5 kV. The mean pore size of each cryogel was calculated by measuring the width of 50 pores using Image J software analysis from SEM images.

#### 2.4.3. Swelling Ratio

The swelling ratio (SR%) of cryogels was monitored in phosphate buffer saline (PBS) at 37 °C, and calculated using Equation (1). Three samples (n = 3, 0.3 cm height, and 1 cm diameter) were used to calculate the average swelling ratio. Firstly, the dry weights of cryogels were recorded. Then, cryogels were immersed in PBS. At certain times, excess PBS from the surface of the cryogels was removed with filter paper, and then the cryogels were weighed. At different time intervals, this process was repeated and the SR% of cryogels was calculated considering time. The swelling ratio/capacity of the cryogels was calculated using the equation below:SR (%) = [(Mf − Mi)/Mi] × 100(1)
where Mi is the initial dry weight of cryogel, Mf is the swollen weight of cryogel, and SR is the swelling ratio.

#### 2.4.4. Contact Angle

The cryogels were cut into 0.3 cm height and 1 cm diameter pieces, and then 5 μL deionized water droplets were added to the surfaces of the samples with a microinjector in an optical tensiometer (Attension Theta, Biolin Scientific, Gothenburg, Sweden). Images were captured at the end of the 5 s after contact of a droplet with the cryogels, by a camera levelled with the surface.

### 2.5. Antibacterial Studies

The antibacterial activities of the cryogels were analyzed according to Pan et al. [34], with slightly modifications. The cryogels were tested against Gram-positive (*Staphylococcus aureus* (*S. aureus*)) and Gram-negative (*Escherichia coli* (*E. coli*)) bacteria. Bacteria were cultured in 10 mL of TSB and incubated at 37 °C for 24 h. Then, this culture for each bacterial species was diluted to the level of 10^8^ CFU/mL, according to McFarland method, at 600 nm. The cryogels were sterilized under UV light for 30 min. Subsequently, 30 mg of sterilized cryogels were mixed with 10 mL bacterial suspensions and placed in a shaker incubator at 37 °C for 2 h, and 24 h. Afterward, the bacterial suspensions including cryogels underwent a series of dilutions, and 100 μL diluted solution was spread evenly on the plate count agar solid medium for inoculation and then incubated at 37 °C for 18 h. Finally, the colonies were counted and recorded.

## 3. Results and Discussion

### 3.1. Microcapsule Formation and Characterization

Essential oils are very sensitive to changes under external factors such as light, temperature, oxygen, and humidity, so they are unstable. The high volatility, hydrophobicity, poor stability, and reactivity of these compounds limit their direct use in longer-term applications. To overcome these limitations, a microencapsulation technique was used to preserve the functional and biological properties of these bioactive compounds while controlling their release [35,36]. Therefore, in this study, we preferred to apply PEO via microencapsulation to increase the antibacteriality of the cryogels and provide a long-lasting antibacterial activity. In other similar studies, this approach has been shown to prolong the release time of the essential oil, thereby preserving long-term antibacterial activity [37,38]. In this context, PEO was encapsulated with the help of the chitosan polymer. The microcapsules were successfully produced in a concentrated NaOH solution. To evaluate the effect of PEO on the characteristics of cryogels, plain chitosan (ChiCap) and PEO-loaded chitosan (PEO@ChiCap) capsules were prepared separately. In Figure 2A,B, the photographs of microcapsules in a wet state, taken in daylight, are shown. The examination of the samples revealed spherical microparticles for both ChiCap and PEO@ChiCap capsules. It was also observed that the particle size of the PEO@ChiCap was smaller than the ChiCap. To support the obtained results, optical microscopy images of the samples were taken (Figure 2C,D). Light microscopy images displayed spherical microcapsules with a dark contrast, which indicates a dense packing of the polymer chains. According to histograms which were created using the particle sizes obtained by processing the microscopy images with Image J, the mean ± standard deviation was calculated to be 2315.06 ± 158.56 µm for ChiCap, and 2019.04 ± 155.77 µm for PEO@ChiCap (Figure 2E,F).

The FTIR spectra of the plain chitosan microcapsules (ChiCap) and PEO-loaded chitosan microcapsules (PEO@ChiCap) are presented in Figure 3 to confirm the successful encapsulation of PEO by chitosan. The bands in the spectral range of 2849–2954 cm^−1^ have previously been ascribed to the presence of menthol and menthone, the two most significant constituents of peppermint oil [39]. The wavenumbers 1057, 1108, and 1243 cm^−1^ correspond to C-O stretching vibrations, whereas the dense absorption bands between 700 and 1455 cm^−1^ result from C-H bond vibrations in the peppermint structure [40]. According to the FTIR findings, PEO was successfully encapsulated in chitosan.

### 3.2. Cryogel Formation and Characteristcs

In this study, four different compositions of gelatin-based cryogels were succesfully fabricated as empty gelatin (GelCryo), PEO-incorporated gelatin (PEO@GelCryo), plain chitosan microcapsule-loaded gelatin cryogel (GelCryo–ChiCap) and PEO-encapsulated chitosan-microcapsule-loaded gelatin cryogel (GelCryo–PEO@ChiCap), as illustrated in Figure 1. The main purpose was to keep the PEO in the cryogel structure without changing the properties of the existing cryogel, even by contributing to its development (surface area, mean pore size, mechanical stability, etc.), and to provide extended release by preventing the burst release during application. Therefore, in this section, the effect of adding PEO@ChiCap on the basic properties of gelatin cryogel is primarily discussed.

#### 3.2.1. Structural Features of the Cryogels

The structural peculiarities of cryogels and cryogels filled with microcapsules were obtained by examining thin sections of these cryogels via SEM. Images in Figure 4 show the surface and inner microstructure of cryogels and the microcapsules located in them at different magnifications (68× and 5000×). In general, as an expected result of the cryogelation process, all samples were exposed an interconnected pore structures consisting of both micro and macro pores. The mean pore size was calculated as 376 ± 183, 171 ± 96, 233 ± 126, and 321 ± 117 µm for GelCryo, PEO@GelCryo, GelCryo–ChiCap, and GelCryo–PEO@ChiCap, respectively. These values are in the range of the pore sizes of gelatin based cryogels produced by other researchers [41,42]. When comparing the GelCryo and PEO@GelCryo samples, the addition of essential oil into the gelatin solution decreased the mean pore sizes of the cryogels. This result can be explained by the fact that the essential oil dispersed in the gelatin, thickening the polymer walls and shrinking the pores. In one of our previous studies and literature search, it was confirmed that a oil–polymer composition directly causes microstructural changes that occur in the network of scaffolds because of the increasing pore wall thickness and decreasing porosity with an increasing oil concentration [10]. Researchers found similar results due to the oil incorporation, which changes the pore morphology of the chitosan scaffolds and polyvinylpyrrolidone/gelatin nanofibers, respectively [43,44]. However, in the samples where the essential oil was encapsulated with chitosan and added to the gelatin (GelCryo–PEO@ChiCap), it was observed that the oil did not disperse in the gelatin solution because it was encapsulated, preserving the existing pore diameter.

Due to the hydrophobic and immiscible properties of the essential oil, its particles can be seen on the material surface in SEM images. When Figure 4F is examined, the direct addition of PEO to the gelatin cryogel structure resulted in small holes on the polymer wall due to the evaporation of oil and a relatively rough surface compared to the wall morphology of GelCryo (Figure 4E). Bubble-like holes may have formed as a result of the evaporation of PEO, a volatile substance, during the pre-treatment of the oil-containing polymer solution before placing it in the freezer, or until the material begins to stabilize in a solid state upon freezing. A similar result was also found by Rezaei et al., whereby they reported that the addition of the saqez essential oil into polyvinyl alcohol/chitosan hydrogels caused identical morphology, like the one found in our study. They attributed the formation of macro pores on the polymer surface to the rapid evaporation of the essential oil [45]. Looking at the cryogel structure loaded with PEO-encapsulated chitosan microcapsules (GelCryo–PEO@ChiCap), as an oil-containing sample, stigma-like structures were observed on the pore wall surfaces (Figure 4H). With this image, compared with GelCryo–ChiCap (Figure 4G), it can be said that the oil is embedded in the chitosan polymer chains.

The cryogel that draws attention in SEM images and that we will focus on in the next stages of our study is GelCryo–PEO@ChiCap, presented in Figure 4D. It is interesting that the PEO@ChiCap, which is produced at room temperature, is added to the gelatin solution and produced together under cryogenic conditions, displaying a highly porous structure in the microcapsules besides the gel structure. Similarly, ChiCap microcapsules were entrapped in the cryogel and exhibited a porous structure in the cross-section, but spread out in the gelatin solution without retaining their spherical shape (Figure 4C, indicated by green hollows).

The highly porous structure exposed as a result of adding PEO-loaded microcapsules (which we prepared at room temperature and washed until the pH was neutral) to the gelatin solution and co-produced under cryogenic conditions, is interesting.

#### 3.2.2. Chemical Features of the Cryogels

The crosslinking of gelatin in the existence of glutaraldehyde, and the presence of PEO, ChiCap, and PEO@ChiCap were analyzed via ATR–FTIR. The common denominator of all gels is the crosslinking of gelatin with glutaraldehyde under cryogenic conditions. It is well known that the crosslinking reaction takes place between the aldehyde functional groups of glutaraldehyde and the unprotonated free amino groups (-NH_2_) of the gelatin in the protein structure, proceeding beyond the stage of a simple Schiff base formation [46].

The characteristic peaks of gelatin and peaks related with the crosslinking of gelatin and glutaraldehyde were successfully determined in all samples (Figure 5). A broad band was observed in the range of 3100–3400 cm^−1^, which corresponds to the stretching vibrations of the non-bonding O-H and N-H functional group presence in the gelatin structure. As the two major peaks which are typical for gelatin, the absorption band at 1639 cm^−1^ was the characteristic peak of amide I (C=O stretching vibration), the band at 1531 cm^−1^ was the characteristic of amide II (N-H bending vibration), and the peak at 1239 cm^−1^ was due to the amide III (C-N stretch plus N-H in phase bending) [47,48]. One of these peaks, amide I at 1639 cm^−1^, was due to the C=N stretching vibration of the imine group of the Schiff base, which confirmed the crosslinking formation between gelatin chains by glutaraldehyde.

For PEO-incorporated cryogels, the presence of PEO in the gels (for PEO@GelCryo and GelCryo–PEO@ChiCap) was also demonstrated in FTIR spectra. In previous studies, bands in the spectral range of 2849–2954 cm^−1^ have been attributed to the presence of menthol and menthone, the most important components of peppermint oil [39,49]. In our results, we also observed similar prominent bands in the same spectral region of both PEO@GelCryo and GelCryo–PEO@ChiCap cryogel samples. The absence of any new peaks, other than the components of the oil, with the PEO additive indicates that the essential oil and carrier materials (both chitosan and gelatin) form a simple mixture rather than a strong chemical interaction. While evaluating the release of PEO from the carrier structure for further studies, it has been observed that the incorporation of PEO through encapsulation into the gel structure would be more advantageous than a direct addition.

#### 3.2.3. Swelling Ratio of the Cryogels

To evaluate the swelling behavior properties of cryogels, we first evaluated the water uptake capacity of cryogels (Figure 6). The SR values of all samples raised in time until the swelling equilibrium was attained. The cryogels displayed a rapid swelling behavior in the first 5 min, then this increment continued and reached an equilibrium after approximately 2 h. The highest values of the equilibrium SR were obtained for the GelCryo–ChiCap and GelCryo–PEO@ChiCap samples. This result indicated that the presence and composition of chitosan microcapsules added to the gelatin structure had a significant effect on this behavior. As presented in the SEM images (Figure 4D), the chitosan microcapsules exhibited an extremely high porosity. Thus, by adding the porosity of chitosan microcapsules to the existing gelatin porosity, a significant increase in the surface area may have occurred. Previous research has shown that increasing porosity increases the contact surface area between the cryogel and the medium, resulting in an increased swelling rate [50].

#### 3.2.4. Contact Angle of the Cryogels

The assessment of the surface hydrophobic and hydrophilic properties of the cryogels was conducted through the utilization of water contact angle (θ, degrees) measurements. The water contact angles of GelCryo, PEO@GelCryo, GelCryo–ChiCap, and GelCryo–PEO@ChiCap cryogels were 99.98°, 76.90°, 66.11°, and 70.91°, respectively, as shown in Figure 6. It can be seen that the contact angle for PEO@GelCryo and GelCryo–PEO@ChiCap samples containing PEO is lower than the empty gelatin cryogel (GelCryo). This may be due to the water repelling of the essential oil, which is known to exhibit hydrophobic properties. As described in the study of Arik et al., the presence of oil increased the hydrophilicity and decreased the contact angle [51]. On the other hand, there was a significant reduction in the contact angle of GelCryo–ChiCap (66.11°) compared to GelCryo (99.98°). This may be related to the existence of highly porous microcapsules in the cryogel structure. Although surface chemistry appears to be the most important element influencing structural wettability, changes in porosity may also lead to a decrease in the contact angle [52]. When the researchers investigated the influence of the absorbent substrate’s porosity on the contact angle, they discovered that greater porosity increases fluid penetration and, as a result, the contact angle lowers [53].

### 3.3. Antibacterial Activity of Cryogels

Figure 7 shows the *S. aureus* and *E. coli* growth curves in relation to the aerogels. The Gelcryo displayed no growth inhibition effect, whereas the crygels loaded with PEO and chitosan microcapsules exhibited antibacterial efficiency. In the cryogels containing antibacterial agents (PEO and chitosan), the growth of bacterial cultures tended to diminish at 2 h and 24 h, whereas it increased in plain cryogels. Compare to the PEO@GelCryo which had the highest antimicrobial activity at 12.06% against *S. aureus* and 7.77% against *E. coli* at 2 h, GelCryo–PEO@ChiCap showed the lowest antimicrobial activity, at 0.93% and 1.29% against *S. aureus* and *E. coli*, respectively. However, after 24 h incubation, GelCryo–PEO@ChiCap exhibited the highest inhibition against *S. aureus* and *E. coli*. This result could be related to the control release of PEO encapsulated in microcapsules. Also, cryogels loaded with chitosan microcapsules demonstrated antibacterial activity against *S. aureus* (8.29%) and *E. coli* (5.22%) in growth inhibition for 24 h. Batista et al. [54] formulated novel alginate/chitosan aerogels, and found that the aerogels with a higher content of chitosan showed maximum antibacterial activity.

Essential oils have been studied extensively for their antibacterial properties, and this property has been linked to essential oils’ capacity to permeate the outer membranes and cytoplasmic membranes of bacterial cells, leading the cell structures to break down and become more permeable [55,56]. It is generally recognized that essential oils have a greater inhibitory effect on Gram-positive bacteria compared to Gram-negative bacteria [28,57,58]. Gram-positive and gram-negative bacteria have different cell wall structures, which affect their susceptibility to essential oils. Gram-positive bacteria have a thick peptidoglycan layer in their cell walls, which is more porous and accessible to essential oil components. This makes Gram-positive bacteria more susceptible to the antimicrobial properties of essential oils. Essential oils can easily penetrate the cell walls of Gram-positive bacteria, disrupt their membrane integrity, and inhibit their growth and reproduction. The presence of an outer membrane composed of lipopolysaccharides in Gram-negative bacteria reduces their susceptibility to essential oils and makes them less susceptible to inhibition compared to Gram-positive bacteria [59,60,61].

The results of the present study demonstrate that microencapsulated antibacterial compounds loaded into cryogels can be a promising alternative for delivering higher antibacterial activity long-term. Furthermore, cryogels can be designed to have specific properties, such as pore size, surface chemistry, and mechanical strength, to optimize antibacterial activity. The choice of gel precursors and crosslinking agents during cryogel synthesis can be tailored to achieve the desired characteristics. Further research is needed to fully understand the mechanisms of action and potential applications of essential oils as antimicrobial agents.

## 4. Conclusions

The incorporation of bioactive compounds, such as essential oils and plant extracts, into gel matrices has gained significant attention for its potential applications in various fields, including the food industry, drug delivery, and tissue engineering. The main purpose is to provide antibacterial properties to existing systems. The primary challenge observed in this context pertains to the rapid and burst release of the incorporated bioactive substance from the material. These findings indicate that essential oils do not elicit a chemical reaction with polymers, but rather manifest only as a physical attachment. For this reason, it is of utmost significance for long-term activity that the oil can be held inside the material. Based on this idea, PEO was microencapsulated in chitosan and incorporated into gelatin cryogels. A crosslinking reaction with glutaraldehyde in the presence of microcapsules resulted in a highly porous structure of both the gelatin matrix and the chitosan microcapsules. As a result, the cryogels’ increased surface area as a result of porosity led to a high rate of water retention and the development of hydrophilic characteristics on their surfaces. GelCryo–PEO@ChiCap was shown to have the highest inhibition against *S. aureus* and *E. coli* after one-day incubation due to PEO, assumed to control the release from microcapsules. This demonstrated that PEO was encapsulated in the chitosan microcapsules. The preliminary results obtained have proven that PEO-encapsulated chitosan-particle-loaded gelatin cryogels can be further evaluated for additional functional applications. Future research should look into the effects of loading these composite gels with different bioactive compounds on their release behavior and the investigation of microbial activities during the release period.

## Figures and Tables

**Figure 1 polymers-15-02782-f001:**
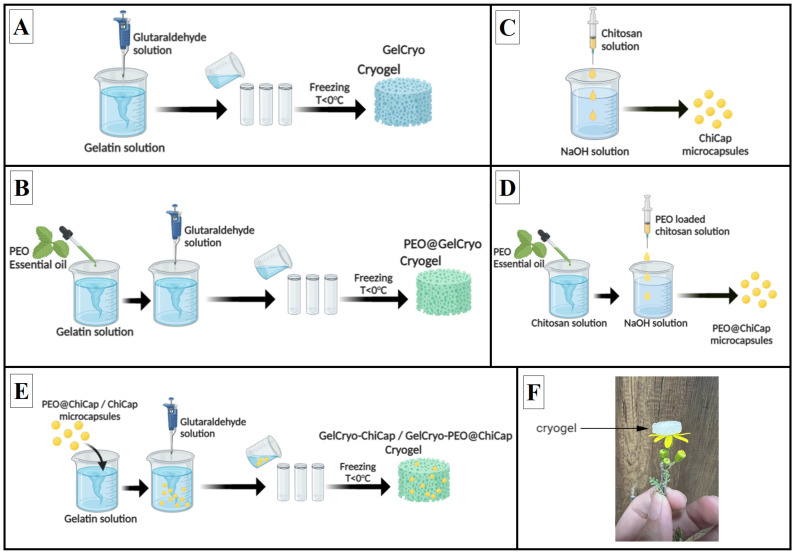
Process steps of cryogel formation. (**A**) Gelatin (GelCryo), (**B**) PEO-incorporated gelatin (PEO@GelCryo) cryogels, (**C**) plain chitosan microcapsules (ChiCap), (**D**) PEO-loaded chitosan microcapsules (PEO@ChiCap), (**E**) ChiCap-loaded cryogel (GelCryo–ChiCap) and PEO@ChiCap-loaded cryogel (GelCryo–PEO@ChiCap), and (**F**) representative photo.

**Figure 2 polymers-15-02782-f002:**
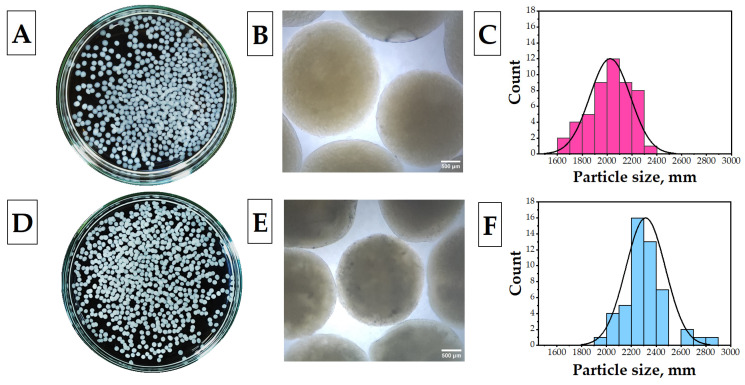
Daylight photography, microscope image, and particle size distribution histograms of the microcapsules. (**A**,**C**,**E**) Plain chitosan microcapsules (ChiCap) and (**B**,**D**,**F**) PEO-loaded chitosan microcapsules (PEO@ChiCap).

**Figure 3 polymers-15-02782-f003:**
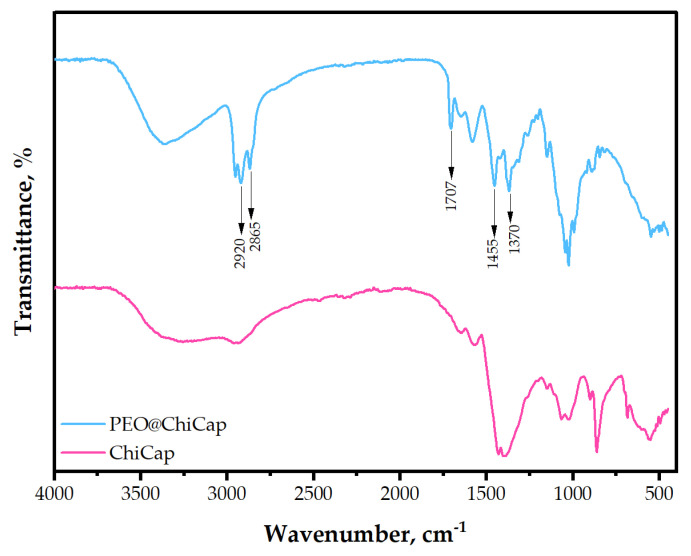
FTIR spectrum of plain chitosan microcapsules (ChiCap) and PEO-loaded chitosan microcapsules (PEO@ChiCap).

**Figure 4 polymers-15-02782-f004:**
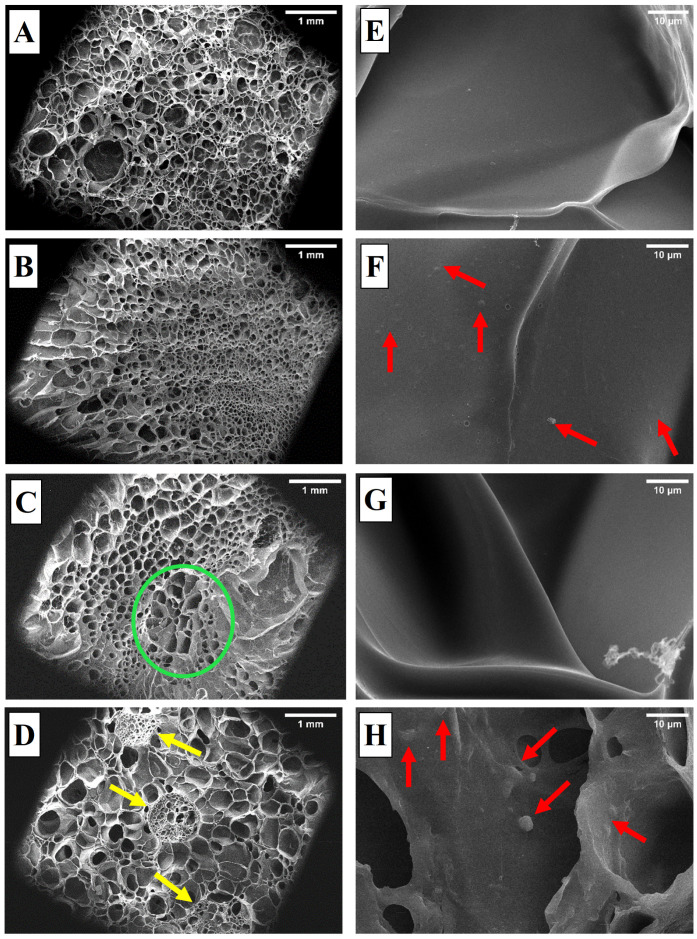
The inner and surface morphology of cryogels obtained using SEM images. (**A**,**E**) GelCryo, (**B**,**F**) PEO@GelCryo, (**C**,**G**) GelCryo–ChiCap, and (**D**,**H**) GelCryo–PEO@ChiCap (yellow arrows indicate microcapsules, green hallows indicate non-stable and spread-out microcapsules, red arrows indicate oil trapped in the cryogel pore wall).

**Figure 5 polymers-15-02782-f005:**
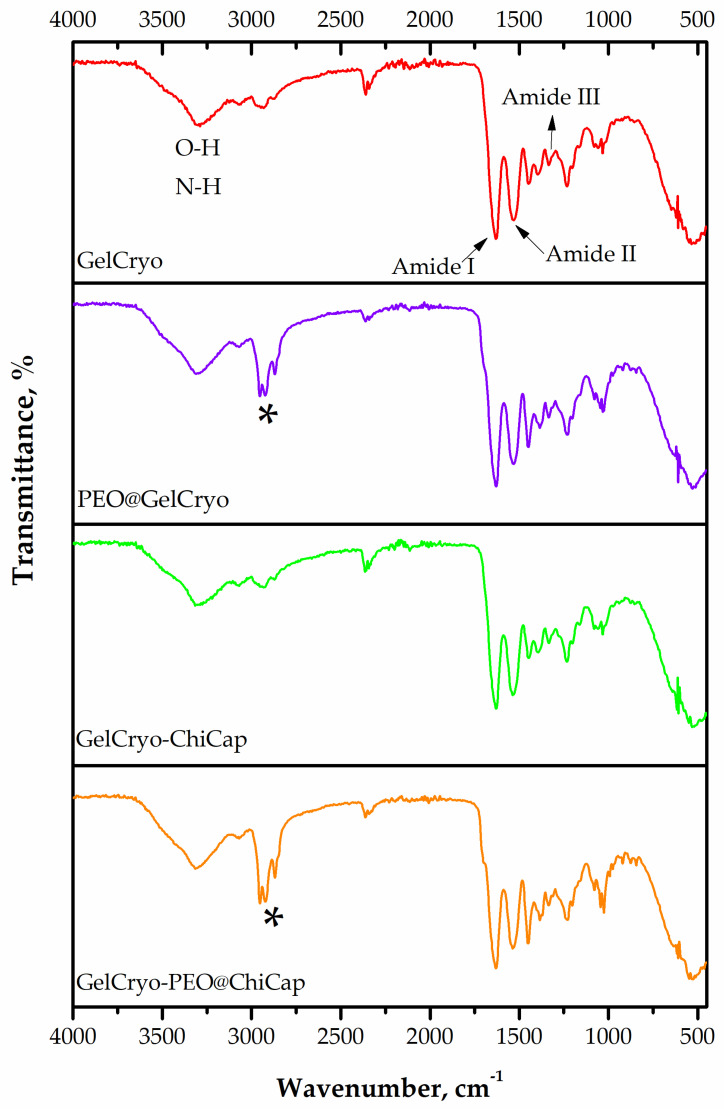
FTIR spectra of GelCryo, PEO@GelCryo, GelCryo–ChiCap, and GelCryo–PEO@ChiCap cryogels (asterisks indicate the presence of PEO).

**Figure 6 polymers-15-02782-f006:**
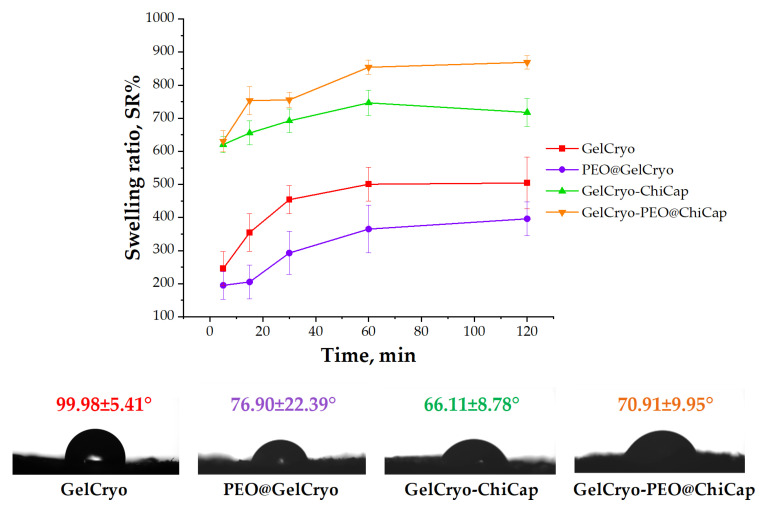
Surface characterization of cryogels by swelling ratio and contact angle measurements. Swelling ratio analysis was performed in water. Contact angle of the cryogel samples was performed with a 2.77 μL mean droplet volume of water.

**Figure 7 polymers-15-02782-f007:**
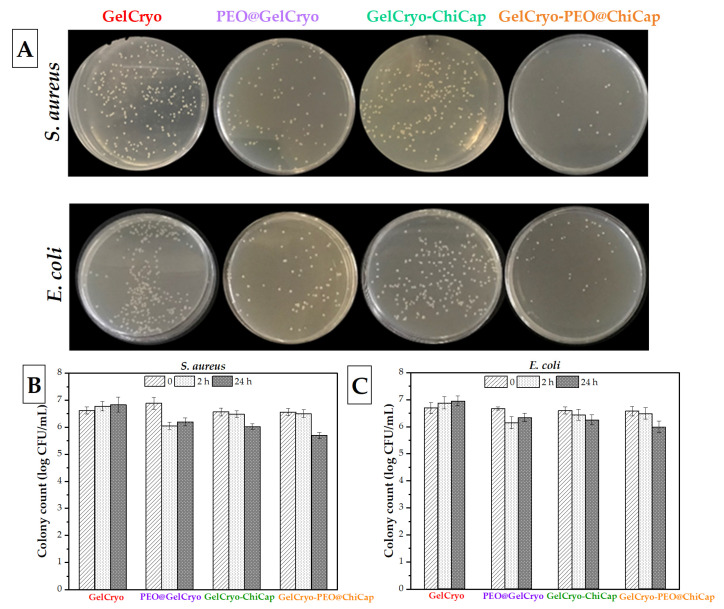
Visual photos of the inhibition properties of the cryogels against (**A**) *S. aureus* and *E. coli* in the petri dishes, and the growth graph of (**B**) *S. aureus* and (**C**) *E. coli* inoculated in cryogels at 0, 2 and 24 h incubation time.

## Data Availability

The data that support the findings of this study are available upon reasonable request from the authors.

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
