# Peer review of "Optimized Peppermint Essential Oil Microcapsules Loaded into Gelatin-Based Cryogels with Enhanced Antimicrobial Activity"

_polymers, 2023, doi:10.3390/polym15132782_

Round 1
Reviewer 1 Report (New Reviewer)
This manuscript discussed the release of peppermint essential oil loaded in gelatine based cryogels. It is a very interesting topic to discuss and the manuscript is well written, however, I still have some comments:
1- How was the crosslinking reaction with glutaraldehyde performed? Was the concentration of glutaraldehyde optimized to achieve the desired porosity and structure of the gelatin matrix and chitosan microcapsules?
2- It is mentioned that GelCryo-PEO@ChiCap exhibited the highest inhibition against S. aureus and E. coli after one-day incubation due to PEO controlled release from microcapsules. Could you elaborate on the release behavior of PEO from the chitosan microcapsules? Was it a sustained release or an initial burst followed by a slower release? and do the authors expected to follow the same profile of the swelling (Figure 6)?.
3- In future studies, as suggested, it would be interesting to investigate the loading of these composite gels with different bioactive agents, do the authors think there is a way to release the EO in a controlled manner?
4- The sentence "So essential oils are more inhibited effect against to gram positive than gram negative bacteria" 452. should be improved to something like "essential oils have a greater inhibitory effect on gram-positive bacteria than on gram-negative bacteria"
Author Response
Dear Editorial Board,
Upon receiving the comments from Reviewers, it was noticed that Reviewers mainly have make constructive and advising comments on the manuscript. Therefore, it was decided to revise manuscript in detail for its publication in the Polymers journal. With the revised version, we believe that manuscript has the required quality and impact to be considered for its publication in the journal.
We would like to express our deepest gratitude for Reviewers’ constructive comments. The comments from Reviewers have been carefully considered and after a detailed study, corresponding changes were made in the manuscript. The revisions and actions for each of the comment and the revised manuscript are enclosed.
We are looking forward to hearing from you.
Reviewer #1
This manuscript discussed the release of peppermint essential oil loaded in gelatine based cryogels. It is a very interesting topic to discuss and the manuscript is well written, however, I still have some comments:
Comment 1: How was the crosslinking reaction with glutaraldehyde performed? Was the concentration of glutaraldehyde optimized to achieve the desired porosity and structure of the gelatin matrix and chitosan microcapsules?
Response of Authors: The concentration of glutaraldehyde as a crosslinking agent is directly related to the pore structure, porosity, water holding capacity and mechanical properties of the materials produced. In our study, the possible reaction between glutaraldehyde and gelatin was explained in section “3.2.2. Chemical Features of the Cryogels”. In addition, regarding the optimization of the concentration used, researchers from our team have optimized the amount of glutaraldehyde according to the polymer type and amount in many previous studies as listed below. Using the experience in these processes, it was not necessary to make a change in the glutaraldehyde ratio in the current study.
https://doi.org/10.1002/app.50337, https://doi.org/10.1177/08853282211048284, https://doi.org/10.1002/pi.6372, https://doi.org/10.1007/s10853-021-06399-8, https://doi.org/10.1080/21691401.2021.2012184
Comment 2: It is mentioned that GelCryo-PEO@ChiCap exhibited the highest inhibition against S. aureus and E. coli after one-day incubation due to PEO controlled release from microcapsules. Could you elaborate on the release behavior of PEO from the chitosan microcapsules? Was it a sustained release or an initial burst followed by a slower release? and do the authors expected to follow the same profile of the swelling (Figure 6)?
Response of Authors:
This result could be related to the control release of PEO encapsulated in microcapsules and it can be assumed follows the same principle of control release.
Comment 3: In future studies, as suggested, it would be interesting to investigate the loading of these composite gels with different bioactive agents, do the authors think there is a way to release the EO in a controlled manner?
Response of Authors: First of all, thank you for supporting. The main problem encountered when using essential oils directly in the material, without encapsulating them with polymers or preparing nano/microemulsions in the presence of surfactant, is the burst release of the essential oil. In fact, the aim of this study was to achieve a time-dependent controlled release of active substance over longer periods of time by embedding the essential oil in the polymer by encapsulation, thus providing a longer-lasting antimicrobial environment. This hypothesis of controlled release behavior is only possible by determining the amount of essential oil that passes into the environment over time with UV-Vis. In our current study, we were able to examine the encapsulation of essential oil and the placement of the produced capsules in a porous material while preserving their shape, and its effect on microbial properties. With your reference to the controlled release behavior, we decided to add to our proposed new study of the use of different essential oils, an examination of the release profiles and release kinetics of these oils. Thank you for your contribution.
Comment 4: The sentence "So essential oils are more inhibited effect against to gram positive than gram negative bacteria" 452. should be improved to something like "essential oils have a greater inhibitory effect on gram-positive bacteria than on gram-negative bacteria"
Response of Authors: We would like thank to Reviewer for his/her valuable comments. Sentence is changed according to comment.
Reviewer 2 Report (Previous Reviewer 3)
In general, the authors significantly improved the manuscript, but did not answer all the questions accurately. For example, the authors respond that they cited the article: 10.3390/molecules27186129. However, this reference is not listed in the bibliography. In addition, it is desirable to double-check the quality of the English language.
Author Response
Dear Editorial Board,
Upon receiving the comments from Reviewers, it was noticed that Reviewers mainly have make constructive and advising comments on the manuscript. Therefore, it was decided to revise manuscript in detail for its publication in the Polymers journal. With the revised version, we believe that manuscript has the required quality and impact to be considered for its publication in the journal.
We would like to express our deepest gratitude for Reviewers’ constructive comments. The comments from Reviewers have been carefully considered and after a detailed study, corresponding changes were made in the manuscript. The revisions and actions for each of the comment and the revised manuscript are enclosed.
We are looking forward to hearing from you.
Comment #1 : In general, the authors significantly improved the manuscript, but did not answer all the questions accurately. For example, the authors respond that they cited the article: 10.3390/molecules27186129. However, this reference is not listed in the bibliography. In addition, it is desirable to double-check the quality of the English language.
Response of Authors: We apologize for the shortcoming. The reference has been added in the revised manual. The quality of the manuscript has been improved.
This manuscript is a resubmission of an earlier submission. The following is a list of the peer review reports and author responses from that submission.
Round 1
Reviewer 1 Report
In this article, the authors systematically evaluated the novel gelatin-based cryogel loaded with peppermint essential oil microcapsules with antimicrobial properties. The description is comprehensive, and the conclusion is significant. Therefore, I recommend this article be published in the Processes after minor revision.
Reviewer 2 Report
In this manuscript Demir, Rusu et.al describe the preparation of glutaraldehyde cross-linked gelatin cryogels loaded with PEO incapsulated in chitosan microcapsules, and these cryogels were tested for antibacterial activity. The idea is interesting, and the growing problem of resistant bacteria nowaday make anything able to contrast them an important research task. However, the manuscript present several issues, and I cannot recommend it for publication in Polymers.
In particular:
-Introduction: the cross-linking of gelatin, that is a key point for the production of these cryogels, is missing, and in the literature there is a plethora of examples.
-introduction: an important work on PEO antibacterial activity is missing (Songklanakarin J. Sci. Technol, 2014, 36.1: 83-87)
-introduction: Antibacterial activity of chitosan has been recently demonstrated (see for example https://doi.org/10.1038/s41598-022-12150-3), such information has to be described, and not only later in ref 42.
-Line 114: "according to previously described methods" references about those methods must be added.
-Line 119: ""put in the freezer quickly to start the crosslinking reaction" it seems that the cross-linking is activated by lowering the temperature, while normally a lower temperature inhibite chemical reactions.
-3.1: there are no analyses that proof the presence of PEO into PEO@ChiCap, maybe FTIR would be useful.
-Line 258: "This result can be explained by the fact that the essential oil dispersed in the gelatin, thickening the polymer walls and shrinking the pores" Are there any study to sopport this affirmation?
-Line 266: "which may have been formed as a result of the rapid evaporation of the essential oil" if the PEO evaporate so quickly why PEO@GelCryo display such long antibacterial activity?
-Line 267: it should be Figure 3F, Figure 3E represent cryogel without PEO.
-Figure 3: it is not sure that those indicated by yellow arrows are chitosan microcapsules, they might be substructures of gelatin itself. More probable are those in Figure 3D, but their dimensions are out of particle size distribution displayed in figure 2, that show no partcles with a size lower than 1.6 mm.
-3.2.2 first paragraph: The cross-linking of proteic substrates with glutaraldehyde is a so known method (first report date back to 1969 https://doi.org/10.1016/0019-2791(69)90178-5), this long description is pointless.
-Figure 4: Are there any free lysine in gelatin? ; Why glutaraldehyde transform into a ketone? ; This ketone react with ammonia? Please take a look at Figure 1 of ref. 35.
-3.2.3: swelling behavior has nothing to do with surface properties, it is a property of the whole material.
-Line 355: from Figure 3D it seems that the areas indicated as chitosan microparticles display smaller pores respect to gelatin. Why this should be the cause of a lower contact angle?
-Line 377: ref. 43 is not appropriate for "Essential oils as sustainable sources of bioactive compounds", this article is focused on plant biomass waste and essential oils are cited only in a phrase when ultrasonic extraction is described. No information on bioactivity of essential oils are present in this work.
-Even if an antibacterial activity of these cryogels has been demonstrated, 24h is a too short time to claim it as "long-lasting".
Reviewer 3 Report
This article is devoted to the production of cryogels with microcapsules of peppermint essential oil. The materials obtained by the authors may have long-acting antimicrobial activity. The article is written at a high level, the amount of data and the subject correspond to the subject and requirements of the journal. There are some points that could be improved:
1. Abstract can be expanded.
2. It is desirable to indicate the component composition of peppermint essential oil, which was used by the authors.
3. Please add more literature comparisons. This will expand the description of the results and lead to greater validity of the conclusions.
4. Figure 3. What is the approximate pore size and type for all studied materials.
5. Please cite: 10.3390/molecules27186129.